

# Recent timescale transition from interannual to decadal variability in January sea ice area over the Bering Sea

Weibo Wang [1,2] Chunsheng Jing [1,2] and Junpeng Zhang [1,2]

1 Ocean Dynamics Lab., Third Institute of Oceanography, Ministry of Natural and Resources, Xiamen 361005, P. R. China.

Fujian Provincial Key Laboratory of Marine Physical and Geological Processes. Xiamen 361005, P.R. China

*Correspondence to:* Weibo Wang (wangwb@tio.org.cn)

**Abstract:** Over the past four decades, the sea ice area (SIA) in the Bering Sea has shifted from interannual to decadal variability, manifested as persistent heavy-ice or light-ice regimes. However, the mechanisms driving this shift remain unclear. This study demonstrates that the initial shift occurs in January and is triggered by the December SIA anomalies. Specifically, December SIA

anomalies induce substantial modifications in localized air-sea heat flux, triggering mesoscale vertical air movements. This process generates localized anticyclonic wind field anomalies during heavy-ice years and anomalous cyclonic wind field anomalies during light-ice years. Subsequently, these mesoscale dynamic processes activate negative feedback in the atmosphere and positive feedbacks in the ocean, which differentially regulate wind divergence and northward heat transport. The former produces out-of-phase variations between December SIA and January SIA increment (ΔSIA), contributing to interannual variability in January SIA,

whereas the latter exhibits significant decadal variability over the past two decades, inducing in-phase changes that amplify decadal-scale signals in sea ice variability. The study emphasizes the critical role of mesoscale ice-atmosphere-ocean coupling processes and their profound impacts on regional oceanic dynamics and sea ice evolution. Given the observed decadal-scale regime shifts in sea ice, of paramount importance and urgency is to assess the implications of sustained heavy/light ice conditions on local ecosystems, indigenous communities, and commercial fisheries.



## 1   Introduction

Bering Sea has experienced a long-term low record of winter sea ice area (SIA) since 2014 (Hunt et al., 2022; Iida et al., 2020; Thoman et al., 2020; Wang et al., 2022, 2023a, 2024a). In the winter of 2018, the maximum SIA plummeted to $1.68 \times 10^5$ km², constituting a mere 30.8% of the historical average (Wang et al., 2023b). This sustained ice loss has triggered profound changes

in regional hydrology, meteorology, ecosystems and even socioeconomic dynamics (Thoman et al., 2020; Wyllie-Echeverria and Wooster, 1998), including weakened seawater stratification (Kinney et al., 2022; O'Leary et al., 2022), delayed spring blooms (Huntington et al., 2020; O'Leary et al., 2022), diminished abundance of large crustacean zooplankton (Belkin and Short, 2023; Hermann et al., 2021; O'Leary et al., 2022; Stabeno and Bell, 2019), and shrinkage or complete disappearance of the cold pool (Kinney et al., 2022; Stabeno and Bell, 2019). Furthermore, reduced SIA in the Bering Sea has been linked to amplified extreme

climate events across mid-and high-latitudes of the Northern Hemisphere, with pronounced impacts on Northeast Asia and North America (Iida et al., 2020; Li and Wang, 2013; Ma and Zhu, 2022; Vihma, 2014; Wu et al., 2009; Zhao et al., 2004; Zhou and Wang, 2008, 2014). Scientists anticipate that the Bering Sea could be ice-free in winter as early as in the next decade (Iida et al., 2020).

Prior to the recent light-ice years, the Bering Sea experienced a decade-long surge in winter SIA. In 2012, the SIA exceeded the

historical average by 55.4% and the highest level since satellite monitoring began in 1979 (Stabeno et al. 2012a,b; Wu and Chen 2016). The climate regime demonstrates distinct decadal oscillations, alternating between several years of relatively extensive sea ice formation and cold summer temperatures (e.g., 2006–2013) and several years of minimal sea ice formation and warm summer temperatures (e.g., 2002–2005, 2014–2021) (Overland et al., 2012; Stabeno et al., 2012a, b; Stevenson and Lauth, 2019; Wang et al., 2022; Yang et al., 2020). This shift contrasts with earlier predominantly interannual/multiyear SIA variability. Prior research

has indicated that hydrological changes in the Bering Sea shelf occur on at least two-time scales—interannual and multi-year (Wyllie-Echeverria and Wooster, 1998). Overland *et al.* (2012) examined the 95-year air temperature record from St. Paul Island in the Bering Sea shelf and determined that decadal warm or cold events are rare and of a random nature. Wu and Chen (2016) pointed out that at the beginning of this century, the Bering Sea SIA in March exhibited significant decadal variation characteristics. According to Yang *et al.* (2020), the decadal variability in March SIA over the Bering Sea began in 2007 and was caused by the

phase-locking of the North Pacific Gyre Oscillation (NPGO) and the Pacific Decadal Oscillation (PDO). According to Wang *et al.* (2022), the decadal variability of January SIA increment (the SIA in January minus the SIA in December of the previous year) in the Bering Sea may have started in 1994 and is closely related to the northward heat transport over the Bering Sea shelf. This interannual-to-decadal transition in SIA over the Bering Sea has taken place and amplified ecological disruptions and modified hemispheric-scale climate patterns.

The winter spatial patterns of sea ice coverage in the Bering Sea have been metaphorically described as a "conveyor belt", wherein ice is transported from northern source to southern sink through an intermediate zone (Li et al., 2014; Niebauer et al., 1999; Pease, 1980; Wang et al., 2024b). Persistent northerly winds drive the ice southward, while ocean heat transport restricts its further southward advance (Brown and Arrigo, 2012; Li et al., 2014; Stabeno et al., 2007; Wang et al., 2022; Zhang et al., 2010). Although previous studies have identified key drivers of Bering Sea ice variability—including atmospheric circulation anomalies (Niebauer,

1980, 1988, 1998; Niebauer et al., 1999; Sasaki and Minobe, 2005), frequent storms (Mesquita et al., 2010; Rodionov et al., 2007; Screen et al., 2011), ice intake from the Arctic Ocean (Babb et al., 2013; Zhang et al., 2010), and poleward oceanic heat transport (Wang et al., 2022, 2023a)—there remains a notable absence of comprehensive discussions on these impact factors. This gap impedes a holistic understanding of sea ice variability in the Bering Sea, making it much more challenging to comprehend the underlying causes of the timescale transition of sea ice change.

In essence, Bering Sea SIA variability are intricately shaped by a blend of thermodynamic and dynamic processes.



Thermodynamically, ice growth and melt are governed by atmosphere-ice and ocean-ice heat exchange. Dynamically, ice convergence/divergence drives local SIA variability. Seasonally, northern Bering Sea ice changes are dominated by atmospheric surface heat flux, while southern ice edge variability is controlled by ice-ocean heat flux (Li et al. 2014). Wind-driven ice transport further modulates spatial patterns (Li et al., 2014; Zhang et al., 2010). In the context of recent Arctic amplification, reduced meridional sea level pressure (SLP) gradients have weakened westerly winds (Cao and Liang, 2018; Dai et al., 2019; Gramling, 2015; Shepherd, 2016), while enhanced northward heat transport through the Bering Strait (Danielson et al., 2014; Woodgate and Aagaard, 2005; Woodgate and Peralta‐Ferriz, 2021) complicates the wind-ice coverage relationship. These substantial changes occur simultaneously with transitions in SIA variability timescales. A comprehensive analysis and thorough examination are requisite to discern the extent to which they function as driving forces in the timescale transition of sea ice. This article endeavours to tackle two core issues: 1. Unravelling the causes of SIA variability at interannual versus decadal timescales; 2. Investigating the variables responsible for the timescale transition in SIA. The primary emphasis of this study centres on the examination of the SIA in the Bering Sea during the month of January, a topic that will be comprehensively expounded upon subsequently. The manuscript is organized into five sections: The initial section provides clarity on the background and motivations, while the second section meticulously delineates the materials and methods employed in the research. The third section delineates the rationale for investigating SIA in January and expounds upon its primary spatiotemporal characteristics. Additionally, it offers an initial elucidation of the factors contributing to the transition of SIA increment from an interannual to a decadal timescale in January. The fourth section systematically unveils the underlying physical mechanisms governing the timescale transition of sea ice in January. The conclusive section summarizes the key findings and contributions of this article.

## 2    Materials and methods

### 2.1    Materials

We utilize monthly mean NASA-team satellite sea ice concentration (SIC) data derived from the Scanning Multichannel Microwave Radiometer (SMMR), Special Sensor Microwave/Imager (SSM/I), and Special Sensor Microwave Imager/Sounder (SSMIS) (Bootstrap Sea Ice Concentrations from Nimbus-7 SMMR and DMSP SSM/I-SSMIS, Version 2 | National Snow and Ice Data Center). The SIC data is obtained at a spatial resolution of 25 km. To calculate SIA, grid-cell area-weighted SIC values were spatially integrated across the area of interest (51-66°N, 165-205°E). We then derived year-to-year January SIA increment (ΔSIA) and constructed a continuous time series spanning 1979-2023. We also utilized Hadley Centre Sea Ice and Sea Surface Temperature dataset (HADISST) (Rayner et al., 2003) as supplementary validation to confirm the robustness of timescale shift in the January ΔSIA from interannual to decadal variability.

We applied Empirical Orthogonal Function (EOF) analysis to extract the dominant spatial patterns (EOFs) and their corresponding time series (PCs). To quantify the relative importance of the first two EOF modes in driving January ΔSIA variability, we compared their principal component ($\frac{PC^2(k)}{\lambda_k}$, where $\lambda_k$ denotes the eigenvalue). The Maximum Overlap Discrete Wavelet Transform (MODWT) was implemented to decompose the time series of January ΔSIA into multiscale components, enabling explicit identification of timescale transitions in sea ice variability. We systematically analysed the linkages between atmospheric processes in December and concurrent SIA through composite analysis and regression analysis. Causal relationships were quantified using the information-flow theory proposed by Liang (2014), while the impacts of December wind field anomalies and oceanic heat flux on subsequent ΔSIA were assessed through lagged correlation analysis.

We employed the satellite products and reanalysis data to reconstruct the surface currents within the Bering Sea. The key datasets



include: 1) the NOAA optimum interpolation SST (OISST) product at 0.25° resolution, as expounded upon by Reynolds et al. (2007); and 2) the Topex/Poseidon and European Remote-sensing Satellite (ERS) altimetric dynamic topography products, also on

a 0.25°×0.25° grid, as made available by AVISO, (2012) (https://www.aviso.altimetry.fr/). Additionally, atmospheric variables—surface air temperature (SAT), omega, sea level pressure (SLP), wind vector at 10 m, latent heat net flux, net longwave radiation flux net shortwave radiation flux and sensible heat net flux—were obtained from the National Centers for Environmental Prediction/Department of Energy Atmospheric Model Intercomparison Project (NCEP/DOE AMIP-II) reanalysis (Kanamitsu et al., 2002). The net air-sea heat flux is computed as the summation of four individual components with upward fluxes defined as

positive (ocean heat loss). To validate robustness, we conducted additional verification using ERA5 reanalysis data provided by ECMWF (Figure S1-S4). Our validation process demonstrated that the observational results obtained from both datasets are consistent. In this study, we primarily focus on the results derived from NCEP/DOE reanalysis data, while the complementary findings from ERA5 are available in supplementary materials.

In order to investigate SLP responses to extreme sea ice conditions in December, we analysed the Polar Amplification

Intercomparison Project (PAMIP) experiments under CMIP6 using the CESM2 model (Danabasoglu, 2019) . Specifically. three tier-1 time slice experiments were evaluated: (1) pdSST-pdSIC, prescribed 1979-2008 climatological SIC; (2) pdSST-piArcSIC, pre-industrial Arctic SIC representing extreme heavy-ice years (Figure 1b), and (3) pdSST-futSIC, RCP8.5-projected Arctic SIC representing extreme light-ice years (Figure 1a). All experiments used fixed climatological SST with 2000/04/01 initial conditions, running 14 months (2-month spin-up discarded). December–January outputs from the final 12 months were analysed to quantify

SLP anomalies induced by contrasting sea ice states.

In an effort to further expand our analysis, we have incorporated surface and bottom water temperature data obtained from in situ observational records. These data were collected during bottom trawl surveys conducted by the NOAA/AFSC/RACE's Groundfish Assessment Program in the eastern Bering Sea. We accessed this valuable dataset through the technical report published by Rohan et al. (2022). By including these water temperature measurements, we aim to provide a more comprehensive understanding of the

timescale transition in the entire marine environment of the Bering Sea.

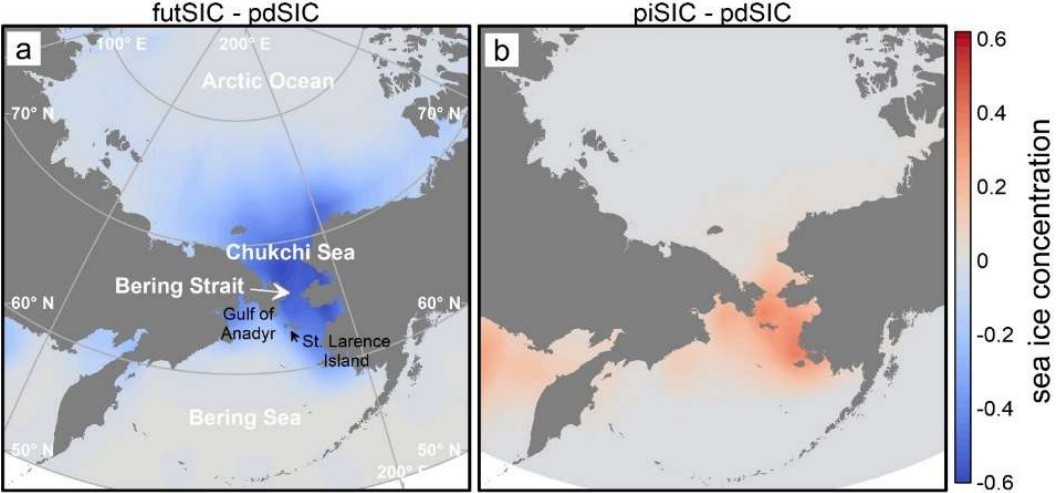

**Figure 1. Comparison of sea ice concentration anomalies between (a) future (futSIC) and (b) pre-industrial (piSIC) climate scenarios from CESM2 simulations. Anomalies are calculated relative to present-day climatological mean SIC (pdSIC).**



### 2.2 Methods

**a) Sea ice area increment (ΔSIA) and maximum SIA**

In this study, we focus on the January ΔSIA to assess the influence of atmospheric or oceanic forcing from the preceding month on sea ice variability. The SIC change is calculated as the difference between the current and preceding monthly SIC values for each grid cell. The January ΔSIA is then derived by applying area-weighting to grid cell-level SIC changes, followed by spatial integration across the region of interest (51-66°N,165-205°E). Anomalies are obtained by subtracting the climatological mean of

the corresponding month. Through comparison of monthly SIA values, the annual maximum SIA in the Bering Sea is identified as the peak value within annual observational record.

**b) Maximal Overlap Discrete Wavelet Transform (MODWT) method**

The Maximum Overlap Discrete Wavelet Transform (MODWT) is utilized to perform the multiresolution analysis of January ΔSIA. As an extension of the Discrete Wavelet Transform (DWT) commonly used in signal processing, MODWT operates as a time-

invariant transform (Walden and Cristan 1998). Distinct from DWT, MODWT, devoid of a downsampling process, minimizes the risk of input signal loss. An advantageous feature of MODWT is its lack of stringent data length requirements, enhancing low-frequency information in multiresolution decomposition. Consequently, the MODWT methodology is applied to decompose the January ΔSIA time series into interannual (1-3 yr), multi-year (4-8 yr), and decadal scales (8-16 yr). To quantify the dominance of variability at each scale, the study calculates relative energy contributions at distinct timescales.

**c) Composite analysis**

Following the Empirical Orthogonal Function (EOF) analysis of the January ΔSIA, the observational period is categorized into five ice-condition classes based on the normalized December SIA ($nSIA_{12}$): normal year (NM, $-0.5 \leq nSIA_{12} \leq 0.5$), heavy-ice year (HI, $0.5 < nSIA_{12} \leq 1$), light-ice year (LI, $-1 \leq nSIA_{12} < -0.5$), extreme heavy-ice year (EHI, $nSIA_{12} > 1$), and extreme light-ice year (ELI, $nSIA_{12} < -1$). Composite differences, encompassing variables such as SIC, net air-sea heat flux, SST, SAT, and wind

vectors, were then derived for each category. Further details can be found in Section 3.2.

**d) Oceanic northward heat transport**

In this study, surface wind vectors and oceanic northward heat transport (*NHT*) are employed to characterize atmospheric and oceanic forcing factors, respectively. We systematically explores their association with the January ΔSIA across various timescales. The formula for computing *NHT* is delineated as follows:

$$NHT = \int_{\lambda_1}^{\lambda_2} \int_{-H_{MLD}}^{0} \rho c_p V T r \cos\theta \, dz \, d\lambda \qquad (2)$$

where $\rho = 1022.95$ kg m$^{-3}$ is the density of seawater; $c_p = 3900$ J kg$^{-1}$°C$^{-1}$ is the specific heat capacity of seawater; $V$ is the meridional surface current; $r = 6371$km is the radius of the Earth; $\theta$ is the latitude, and $\lambda$ is the longitude. In this study, $\rho$, $c_p$, and $r$ are constant, which means that *NHT* is proportional to $VTH_{MLD}\cos\theta$. $T$ is the SST derived from the mean temperature within the depth $H_{MLD}$ (the mixed layer depth).

Prior to calculating *NHT*, the sea surface velocity vector ($\vec{V}(U,V)$) must be determined. It is approximated as the sum of surface geostrophic current ($\vec{V}_{ge}$) and the wind-driven Ekman velocity ($\vec{V}_{ek}$). We employed sea surface dynamic height and wind vectors data to compute surface currents, with the calculation equation articulated as follows:

$$\vec{V_{ek}} = \frac{1}{\rho_0 f}(\tau_y, -\tau_x) \, and \, \vec{\tau} = \rho_0 C_D |\vec{u_s}| \vec{u_s} \qquad (3)$$



$$\vec{V}_{ge} = \frac{g}{f}\left(-\frac{\partial h}{\partial y}, \frac{\partial h}{\partial x}\right) \tag{4}$$

where $\rho_0 = 1.25\ kg\ m^{-3}$ is the air density, $C_D = 0.00125$ is the drag coefficient, $f$ is the Coriolis parameter, $g = 9.8$ m/s$^2$ is the acceleration of gravity, and $h$ is the dynamic topography. The subscripts '$x$' and '$y$' of $\tau$ denote the zonal and meridional directions, respectively. $\vec{u_s}$ is obtained, as described above, from the wind data recorded at 10 m above sea level. This methodology has been extensively validated in oceanic dynamical studies (Dohan and Maximenko, 2010; Sudre and Morrow, 2008; Wang et al., 2024b). Wang et al. (2024a) offers a comprehensive evaluation of the uncertainty associated with ocean heat

transport using this approach.

To unravel the causal links between SIA and atmospheric processes, we implement two complementary approaches: moving-window correlation coefficient (MCC) and Liang-Kleeman information flow analysis. A two-sided Student's t-test was employed to evaluate the statistical significance of the MCC, ensuring a rigorous assessment of the results. The degrees of freedoms is $N_{eff} - 2$. $N_{eff} = N/(1 + 2\sum_{i=1}^{N-1}\frac{N-i}{N}\rho_{x,i}\rho_{y,i})$ is the number of effective degrees of freedoms of the combined dataset, where $N$

denotes the sample size and $\rho_{x,i}$ is the auto-correlation of time series $x$ with lag $i$ (Bayley and Hammersley, 1946). Here, our primary focus lies in the Liang-Kleeman information flow,

**e)   Liang-Kleeman information flow**

Liang and Kleeman have pioneered the introduction of the information flow method as a means to unveil causal relationships embedded within time series data (Liang, 2014). For two given series, $X_1$ and $X_2$, the rate of information flow (expressed in units

of nats per unit time) from the latter to the former is defined as follows:

$$T_{2\rightarrow1} = \frac{C_{11}C_{12}C_{2,d1} - C_{12}^2 C_{1,d1}}{C_{11}^2 C_{22} - C_{11}C_{12}^2} \tag{5}$$

where $C_{ij}$ is the sample covariance between $X_i$ and $X_j$, $C_{i,dj}$ the covariance between $X_i$ and $\dot{X}_j$, and $\dot{X}_j$ is the difference approximation of $\frac{dX_j}{dt}$ using the Euler forward scheme:

$$\dot{X}_{j,n} = \frac{X_{j,n+k} - X_{j,n}}{k\Delta t} \tag{6}$$

where $k \geq 1$ but should not be too large to ensure precision. Practically, a comparison can first be made between the results with $k = 1$ and $k = 2$. If the results are qualitatively different, then $k = 1$ should be discarded. When $T_{2\rightarrow1}$ is significantly different from 0, $X_2$ has an influence on $X_1$, while if $T_{2\rightarrow1} = 0$ there is no influence. A positive $T_{2\rightarrow1}$ means that $X_2$ makes $X_1$ more uncertain, while a negative value indicates that $X_2$ tends to stabilize $X_1$. In this study, the Liang-Kleeman information flow methodology is employed to systematically investigate the causal relationship between concurrent sea ice and atmospheric dynamics.




## 3 Results

### 3.1 Onset of timescale transition of SIA

Several studies have definitively substantiated the timescale transition of sea ice in the Bering Sea from interannual to decadal variability, with a particular emphasis on the maximum SIA. Figure 2 illustrates the time series of the SIA anomaly in December (SIA$_{12}$) and the maximum SIA along with their first three Intrinsic Mode Functions (IMFs). The variability of SIA and its ΔSIA for the months of November to May can be found in Figure S5 of the supplementary materials. A careful examination reveals that the SIA anomalies in December manifest no discernible augmentation in decadal-scale signals (Figure 2a-d). Conversely, the decadal variability of the maximum SIA in the Bering Sea (Figure 2e-h) has experienced substantial amplification since 1995. In conjunction with the timescale transition from interannual to decadal variability, analogous features in the mean temperature of surface and bottom seawater during spring are observed in Figure 2k and 2l, as well as the Cold Pool Index (Figure 2j)—defined as the proportion of the Bering Sea southeastern shelf with a bottom water temperature under 2°C in winter. This observation underscores the presence of a timescale transition within the integrated multi-layer system of the Bering Sea, encompassing from the hydrological environment to the ecosystem, and extending beyond the domain of sea ice. A comprehensive analysis of the time series from November to February reveals that January exhibits the earliest timescale transition (Figure S5), which is strongly associated with the largest ΔSIA in January compared to other months (Wang et al., 2022). This association directly motivates the study's specific focus on the January ΔSIA.

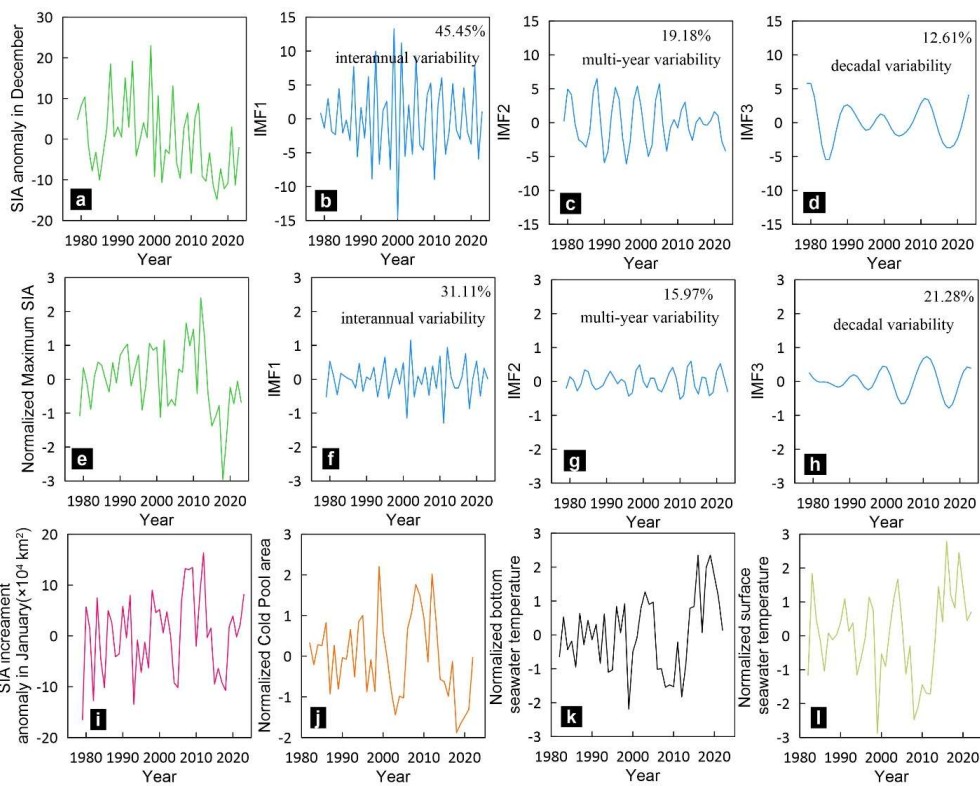

**Figure 2. Time series of the SIA anomaly in December (a) and maximum SIA (e) over the Bering Sea, and their first three**





**Intrinsic Mode Functions (IMFs, b, c, d correspond to December SIA anomaly, and f, g, h correspond to maximum SIA)**

**from 1979 to 2023. Panels (i-l) show the time series of January ΔSIA anomalies (i), normalized cold pool area (j), normalized bottom seawater temperature (k) and surface seawater temperature (l) during the spring period spanning 1981-2022. All decomposition terms, with the exception of multi-year and decadal changes in December, have reached the significance test of 95%, as determined by hypothesis testing.**

### 3.2    Characteristics of the timescale transition of the ΔSIA in January

The first three IMFs of ΔSIA in January are shown in Figure 3. IMF1 demonstrates substantial interannual variability, accounting for 43.9% of the total energy (Figure 3a). IMF2 captures multi-year scale variability, constituting a relative energy of 19.8% (Figure 3b). Simultaneously, IMF3 exhibits significant decadal variability, accounting for 18.7% of the total energy (Figure 3c). IMF1 maintained high-amplitude fluctuation until 1993, followed by a drastic decline where its MCC with ΔSIA fell below the 95% confidence threshold (Figure 3d). Conversely, only in the interval of 1990 to 2010 does IMF2 exhibit a high MCC with ΔSIA

(Figure 3e). IMF3 exhibited minimal amplitude pre-2000. However, a dramatic increase in the MCC between IMF3 and ΔSIA is observed post-2000. Notably, following 2000, the correlation coefficient between IMF3 and ΔSIA reached statistical significance ($p < 0.05$) with the value approaching 0.8 (Figure 3f). Concurrently, the HADISST-based assessment for January ΔSIA further demonstrated an increasing signal of decadal-scale variability, corroborating the robustness of this phenomenon (Supplementary S1). The computed MCCs reveal a distinct shift in the dominant timescales of January ΔSIA: interannual scales prevailed during

1980-2000, transitioning to multi-year and decadal scales thereafter. Complementary wavelet analysis (Supplementary S2) also confirms this temporal transition, showing decadal signal intensification post-1995 that aligns with the MODWT-derived IMF3 characteristics.





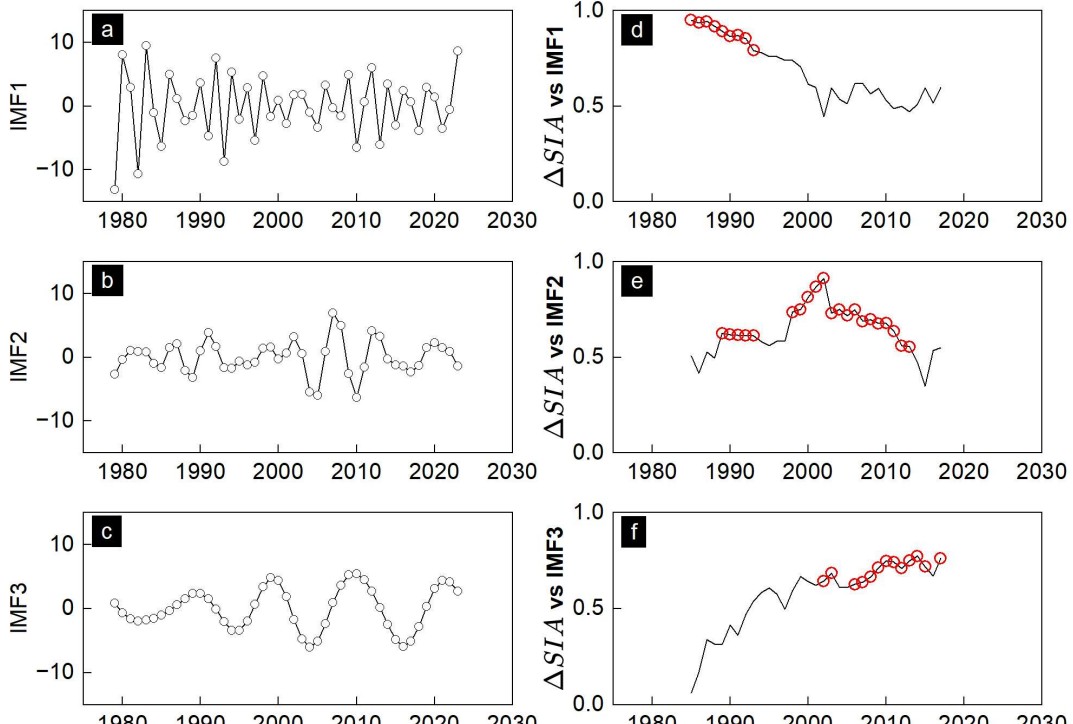

**Figure 3. Multiscale variability of January ΔSIA anomalies characterized through: (a-c) Leading intrinsic mode functions**
**(IMFs) showing variance contributions (numerical labels) to total ΔSIA variability; (d-f) 13-year moving window**
**correlation analysis between IMFs and ΔSIA, with red markers indicating statistically significant level (p < 0.05).**

Wang *et al.* (2022) previously elucidated the spatiotemporal variability of January ΔSIA through the EOF analysis. Building upon
their findings, the MODWT approach was employed to scrutinize the first two PCs, as illustrated in Figure 4. PC1 primarily
displays the multi-year and decadal variability (Figure 4c), accounting for 55.2% of the total energy, whereas PC2 is dominated by
interannual variability (67.2% of the total energy) (Figure 4e). Therefore, the multiyear and decadal signals of ΔSIA primarily
correlate to the spatial pattern of EOF1, in which the area of significant sea ice change includes the region south of the line
connecting Nunivak Island and St. Lawrence Island, as well as the region east of the Gulf of Anadyr (Figure 4a). The interannual
signal corresponds to the spatial pattern of EOF2, with ΔSIA variability concentrated around St. Lawrence Island, extending
eastward to Nunivak Island and westward to the Gulf of Anadyr (Figure 4b). In essence, the temporal evolution illustrated by ΔSIA
from interannual to decadal variability is intricately linked to the shift in dominant spatial patterns from EOF2 to EOF1. Notably,
EOF2 exhibits higher energy than EOF1 between 1983 and 2004, signifying its dominance during this period, followed by a
transition to EOF1 as the leading mode in Figure 4g. To comprehensively understand the mechanisms driving this temporal
transition, it becomes imperative to delve into the governing factors of EOF1 and EOF2.




**Figure 4. Multiscale variability of January sea ice area anomalies (ΔSIA) characterized by: (a-b) EOFs spatial modes (variance explained: EOF1=34.6%, EOF2=26.1%); (c-d) PCs time series with (e-f) their multiscale variability resolved by MODWT into interannual and multiyear+decadal variability; (g) Normalized principal components for the time series of PC1 and PC2. The vertical lines in the panel (g) specifically highlight the years 1983 and 2004, marking the critical points**



**at which the spatial characteristics of the ΔSIA underwent significant transformation.**

While Wang *et al.* (2022) investigated the regulatory factor governing EOF1, a comprehensive analysis of the mechanisms controlling EOF2 and the processes underlying the dominant spatial pattern shift from EOF2 to EOF1 remains lacking. Additionally, the rationale behind the timescale shift, occurring specifically in January as opposed to December, has not been thoroughly examined. Wang *et al.* (2023a) analysed the spatiotemporal change in the December SIA and identified that its dominant pattern, explaining 68% of the variance, is regulated by SLP anomalies. Notably, only a single interannual variation feature is evident in the December SIA time series (Figure 2a), providing evidence that the observed timescale transition in January ΔSIA cannot be solely attributed to early sea ice changes. Instead, this transition likely arises from distinct processes affecting sea ice evolution, a conclusion supported by the demonstrated transformation of its dominant spatial pattern (Figure 4g). A comprehensive examination of the factors driving the observed shift in the dominant spatial patterns of January ΔSIA, and their associated physical mechanisms, will be presented in subsequent sections. However, prior to engaging in an in-depth exploration of these complex topics, it is essential to assess whether a causal relationship exists between December SIA and January ΔSIA. This assessment serves as a foundation for further understanding the intricate dynamics governing the observed SIA changes.

### 3.3 Impact of December SIA on January ΔSIA

A significant negative correlation emerges between $SIA_{12}$ and PC1 after 2000 in Figure 5b. Notably, the magnitude of this correlation increased markedly post-2015, reaching the 95% confidence level. This suggests that the increase in $SIA_{12}$ may function as a precursor to enhanced SIA growth in the subsequent month. Additionally, PC2 exhibits a significant inverse relationship with $SIA_{12}$ prior to 2012. The correlation coefficient in Figure 5c, approaching -1 during the 1990-2010 period, signifies that the increase in $SIA_{12}$ exerts a pronounced inhibitory effect on later SIA expansion. The coexistence of these two opposing effects of $SIA_{12}$ on subsequent ΔSIA implies the existence of distinct mechanistic pathways through which $SIA_{12}$ modulates the subsequent increment in SIA. One pathway displays noticeable interannual variability, whereas the other is associated with long-term decadal-scale changes. When the drivers promote the dominance of the EOF1 spatial pattern, $SIA_{12}$ contributes to fostering subsequent SIA, as exemplified by sea ice changes in the last two decades. Conversely, when EOF2-associated drivers prevail, $SIA_{12}$ impedes the sea ice expansion in later periods, as observed in the sea ice changes of the 1980s and 1990s. Crucially, regardless of whether EOF1 or EOF2 dominates, $SIA_{12}$ acts as a necessary precondition for initiating shifts in ΔSIA's spatial patterns.



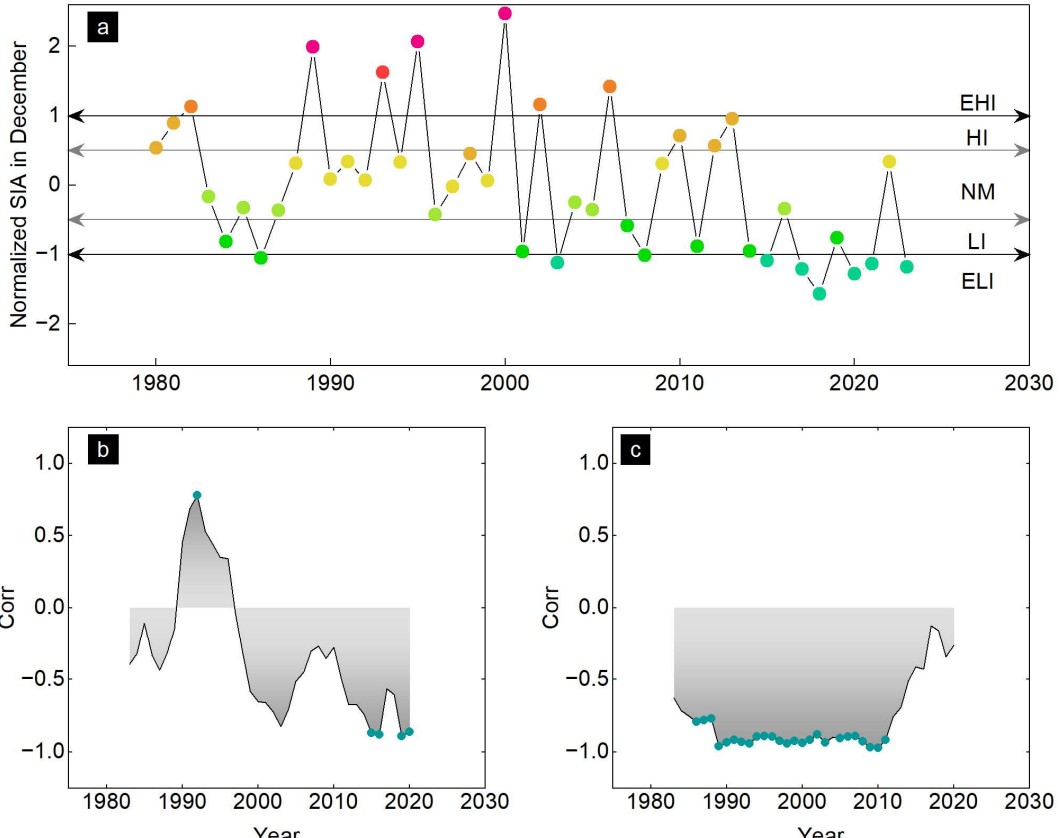

**Figure 5. Statistical linkage between December SIA anomalies and PCs of January ΔSIA. (a) Normalized December SIA; Based on it, observation years are classified into four categories: extreme heavy-ice year (EHI), heavy-ice year (HI), normal year (NM), light-ice year (LI), and extreme light-ice year (ELI). (b) MCC between PC1 and December SIA. (c) MCC between PC2 and December SIA. The green dots within the figures signify correlation that achieve statistical significance at a confidence level of <0.05.**

The analysis suggests a potential causal relationship where December sea ice anomalies exert a forcing on the atmosphere or ocean, thereby modulating subsequent sea ice variability. To rigorously evaluate this hypothesis, we propose classifying the observational years into five distinct regimes based on the normalized $SIA_{12}$—light-ice year (LI), heavy-ice year (HI), normal year (NM), extreme light-ice year (ELI), and extreme heavy-ice year (EHI). This categorization will facilitate systematic investigation of ice-air-sea interactions across different sea ice states, particularly focusing on nonlinear responses under extreme conditions.

### 3.4    Impact of the December SIA on the local net air-sea heat flux

Figure 6 presents composite difference of December SIC, net air-sea heat flux, SST and SAT across four ice-cover regimes relative to the NM years. During LI years, SIC reductions predominantly occur over the eastern Bering Sea shelf, extending toward the Bering Strait (Figure 6a). The most pronounced SIC anomaly (up to 35%) is located near the Bering Strait. The reduced sea ice

coverage facilitates enhanced upward heat flux from the ocean to the atmosphere, with net heat flux anomalies exceeding +80 W/m² (Figure 6e), resulting in a pronounced positive anomaly in SAT, with increases of up to 4 °C near the Bering Strait (Figure 6a). Conversely, HI years exhibit positive SIC anomalies concentrated south of Saint Lawrence Island and in Anadyr Bay, peaking

at 20% (Figure 6b). These ice-surplus conditions correspond to suppressed upward heat flux (negative anomalies of -30 to -50 W/m²) over the anomalous ice cover. However, the associated SAT cooling is relatively modest (-1°C to -3 °C), potentially due to compensatory atmospheric advection or cloud radiative effects (Figure 6b).

In extreme years (EHI and ELI, Figures 6c and 6d), the SIC anomalies exhibit amplified spatial signatures. During ELI years, SIC anomalies are observed across the northern shelf region of the Bering Sea, with a poleward extension into the Chukchi Sea (Figure

6c). The southern Chukchi Sea is characterized by spatially coherent negative anomalies reaching magnitudes of 50%. The reduced ice cover drivers a northward expansion of the positive net air-sea heat flux anomaly (Figure 6g), with localized values exceeding +100 W/m², inducing a SAT positive anomaly of up to 6°C (Figure 6c). During EHI years, persistent high SIC anomalies are anchored over the northern shelf region (Figure 6d). These conditions result in a significant negative anomaly in net air-sea heat flux, peaking at -160 W/m² in Anadyr Bay (Figure 6h). Consequently, the insulation effect of sea ice generates SAT anomalies as

low as -6°C (Figure 6d).





**Figure 6. Composite differences in SIC anomalies (a, b, c, d) and net air-sea heat flux anomalies (e, f, g, h) for the LI year, HI year, ELI year and EHI year in December. Panels (a-d) overlays the contour lines of SAT anomalies at 2 meters, with solid lines indicating positive anomalies and dashed lines representing negative anomalies. The zero line is depicted in red with intervals of 1°C. Similarly, panels (e-h) present the contour lines of SST anomalies, with black lines indicating**



**positive anomalies and gray lines representing negative anomalies.. The zero line is shown in red with intervals of 0.1°C.**

Notably, the relationship between the net air-sea heat flux and SIC anomalies demonstrates distinct nonlinear characteristics. For instance, in LI and HI years, the spatial distribution of SIC anomalies does not align with the regions experiencing significant

changes in net air-sea heat flux. We conducted a statistical analysis to explore the relationship between sea ice and air-sea heat flux in the Bering Sea, with detailed results presented in supplementary materials S1. Overall, in regions where sea ice exhibits obvious changes, the $SIA_{12}$ and net sea-air heat flux display an inverse correlation (Figure 7a), indicating that enhanced local sea-air heat flux is associated with reduced $SIA_{12}$. The substantial positive net air-sea heat flux anomalies directly impact the local atmosphere, resulting in atmospheric warming and promoting upward vertical motion, as depicted in Figure 7b. This dynamical response

subsequently induces horizontal convergence of atmospheric towards the affected region (Figure 7c). Observational evidence from Iida et al. (2020) corroborates the existence of these upward movements in the Bering-Chukchi Sea region, mechanistically linked to sea ice thermodynamic forcing.

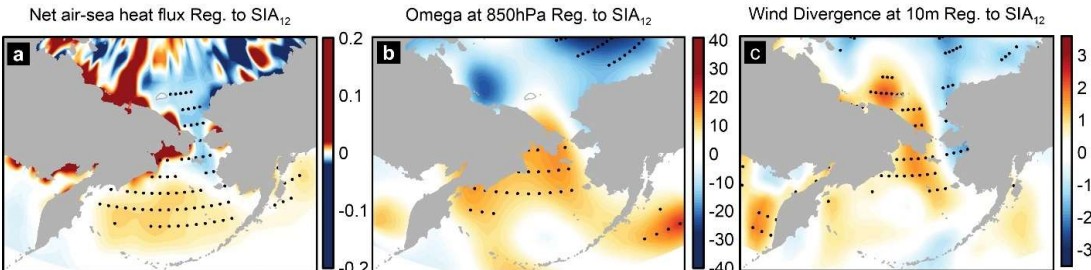

**Figure 7. Regression maps of December net air-sea heat flux (a), omega (b) at 850hpa and wind divergence to the**

**normalized $SIA_{12}$. The black dots within the figures signify correlation that achieve statistical significance at a confidence level of <0.05.**

### 3.5 Adjustment of wind field in December and its impact on the ΔSIA in January

In NM years, SLP exhibit a dipole pattern characterized by two dominant pressure systems: the Siberian High and the Aleutian

Low (AL), which synergistically regulate the climate system over the Bering Sea. The winter robust meridional SLP gradient amplifies geostrophic winds, with prevailing northeasterlies over St. Lawrence Island driving southward sea ice advection in NM years (Figure 8a). Both HI and LI years exhibit positive SLP anomalies (Figures 8b, d). While southerly wind anomalies persist over the northern Bering Sea shelf in both regimes, contrasting wind divergence patterns emerge: LI years exhibit ice-induced surface wind convergence (negative anomalies), whereas HI years show descending air-driven wind divergence (positive anomalies)

associated with extensive ice cover.

The spatial discrepancies in SLP anomalies are even more pronounced during ELI and EHI years. During ELI years, a low-pressure anomaly dominates the southern Bering Sea, attributed to reduced sea ice cover extending northward into the Chukchi Sea (Figure 8c). In areas of diminished sea ice, ascending air induces surface wind convergence and corresponding negative divergence anomalies. In contrast, EHI years feature a pronounced high-pressure anomaly centred over Siberia, with its influence extending

eastward to the western Bering Sea coast (Figure 8d). This anomalous high-pressure governs over the entire Bering Sea, with associated wind divergence propagating eastward across the northern shelf area. Notably, this divergence zone spatially coincides with positive sea ice anomalies, implying thermodynamic coupling between the descending air and positive sea ice anomaly.




Although the distinct SLP patterns observed during ELI and EHI years, their spatial patterns structurally align with the EOF3 mode (Figure 8h), which accounts for 23.6% of the total variance.

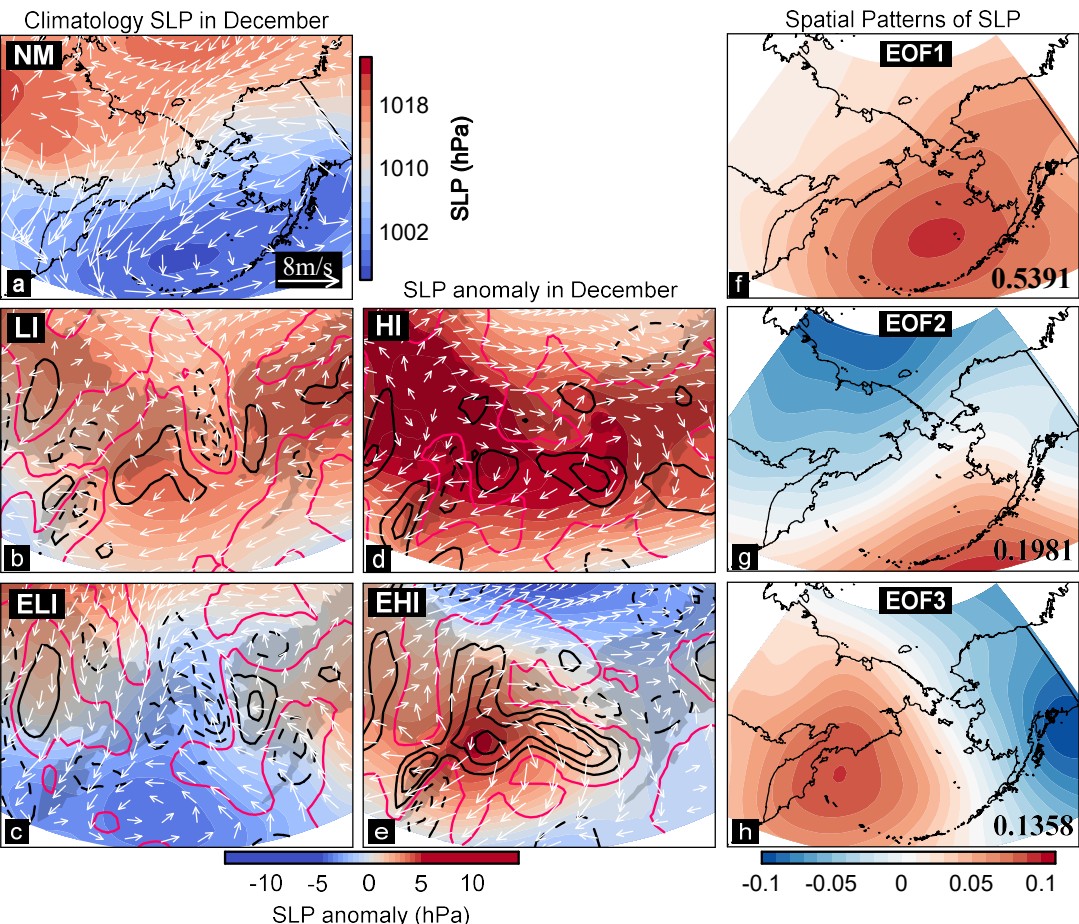

**Figure 8. Response of wind field and SLP to SIA anomalies in December. (a) Climatology SLP (colour shading) and 10 m wind field (white arrows) in NM years. (b, d) Composite differences of SLP and wind vector for the LI and HI years, respectively; (c, e) Composite differences of SLP and wind vector in the ELI and EHI years, respectively. Contours denote wind divergence anomalies (× 10⁻⁵ s⁻¹, solid/dashed lines indicating positive/negative anomalies, respectively. The zero line is shown in red). (f-h) First three spatial modes for the SLP decomposed by the EOF analysis approach, explaining 53.9%, 19.8%, and 13.6% of the total variance.**

As depicted in Figure 7, sea ice-forced vertical atmospheric movement exhibit spatial heterogeneity across the Bering Sea, with significant correlation only collocated with areas of significant sea ice variability. We argued that the substantial alterations in the spatial pattern of SLP over Bering Sea under the four sea ice regimes cannot be exclusively attributed to local ice conditions. We hypothesize these reconfigurations are primarily driven by Arctic-wide climate variability and/or hemispheric-scale atmospheric dynamics. It is essential to underscore that this study specifically targets mesoscale SLP variations within the Bering Sea northern



shelf, rather than basin-scale pressure adjustments. Both LI and HI regimes displayed basin-wide positive SLP anomalies, with the spatial pattern of wind field anomalies maintaining fundamental consistency despite localized discrepancies. Notably, ELI and EHI

events manifested antiphase SLP configurations. In ELI years, the negative wind divergence anomaly extended unusually into the Chukchi Sea, whereas EHI years featured northward expansion of positive anomalies toward the eastern coast of the Bering Sea. These mesoscale signatures demonstrate tight coupling with regional sea ice cover. Mesoscale atmospheric perturbations can drive subsequent adjustments in sea ice conditions. During LI/ELI years, anomalous cyclonic circulation induced by ascending air near Saint Lawrence Island promotes sea ice divergence through wind-driven forcing, thereby enhancing subsequent SIA. Conversely,

During HI/EHI years, sinking-induced anticyclonic flow facilitates sea ice convergence, causing a decrease in SIA. A statistically significant negative correlation emerges between surface wind divergence and PC2 derived from the January ΔSIA over the southern St. Lawrence Island (Figure 9a).

Additionally, wind vectors during the ELI and EHI years exhibit distinct directional disparities as well. In ELI years, the prevailing low-pressure anomaly drives southeasterly wind anomalies across St. Lawrence Island, enhancing Ekman-driven heat advection

northward. Conversely, EHI events feature a high-pressure anomaly that steers predominant westerly anomalies, suppressing meridional heat transport. This disparity in northward heat transport directly modulates subsequent sea ice evolution, as evidenced by a strong positive correlation between northward heat transport and the PC1 from the January ΔSIA over the southern St. Lawrence Island (Figure 9b). Recent mechanistic analyses by Wang et al. (2022) utilizing mixed-layer heat budget diagnostic have quantitatively confirmed the role of December wind fields in governing warm transport variability in the Bering Sea northern

shelf. Their findings corroborated the significant influence of wind fields on the modulation of northward heat transport in the region.

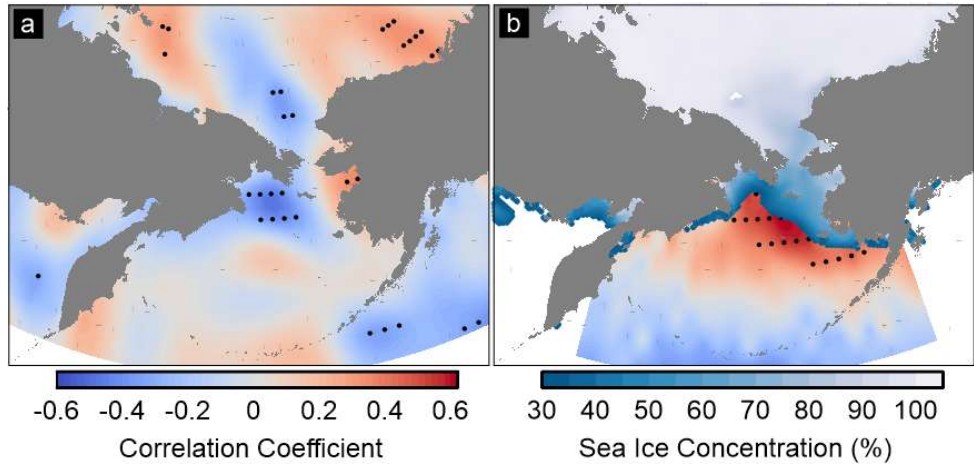

**Figure 9. Linkages between January ΔSIA and atmosphere-oceanic processes. (a) Spatial correlation between the computed PC2 from the January ΔSIA and sea surface wind divergence from 1979 to 2023. (b) Correlation of PC1**

**derived from January ΔSIA and December northward heat transport. Black dots in both panels indicate statistical significance at a confidence level of <0.05. The colour shading in Panel (b) shows the climatological SIC in December over the Bering Sea.**

Figure 10 illustrates the intricate process by which December SIA influence the January SIA, mediated through two dominant

mechanisms. First, an atmosphere-driven negative feedback process: when December sea ice exhibits a positive (negative) anomaly,





the resultant enhanced (reduced) wind field divergence reduces (increases) SIA growth in the subsequent month. This resulting sea ice pattern aligns spatially with the EOF2 mode in January ΔSIA, confirming atmospheric forcing as a primary driver of interannual variability in January ΔSIA. Second, an ocean-driven positive feedback process: A positive (negative) December SIA anomaly induces local anomalous anticyclonic (cyclonic) wind fields. These wind anomalies, in turn, inhibit (facilitate) northward heat 385 transport, thereby enhancing (limiting) SIA growth in the subsequent month. This mechanism corresponds to the EOF1 mode of January ΔSIA and reflects decadal-scale sea ice variability.

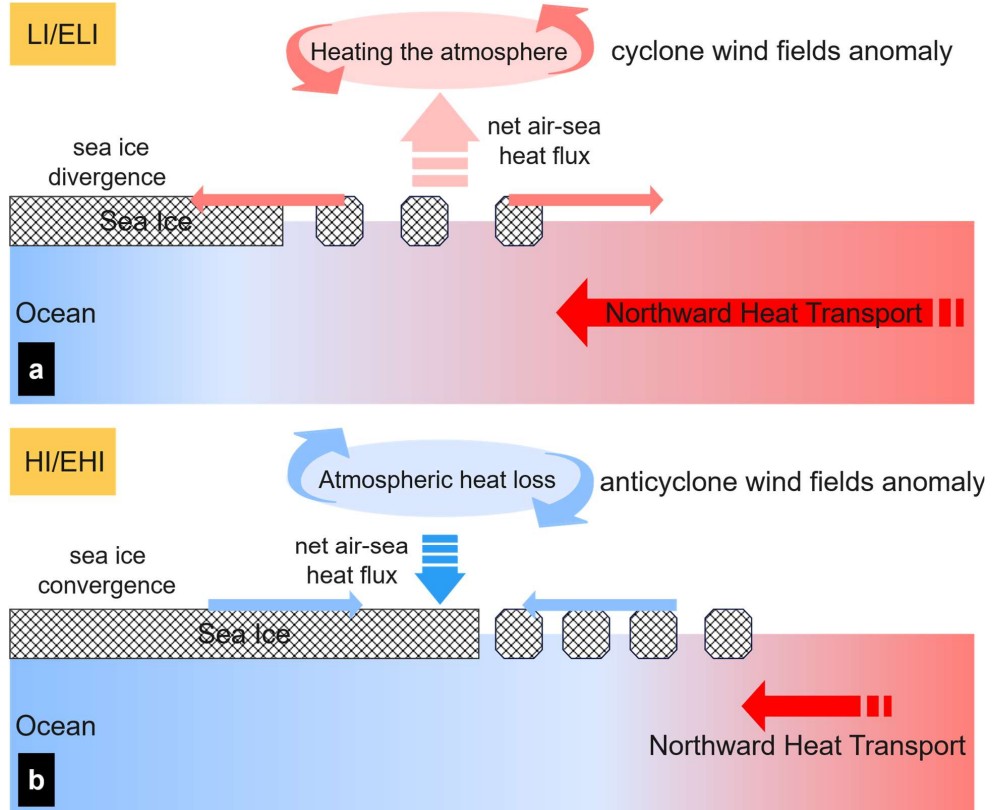

**Figure 10. Mechanistic pathways of atmosphere-ice-ocean interactions driving SIA anomalies. In Panel (a), a vertical-** 390 **meridional view shows the process observed in LI/ELI years, wherein ascending air, induced by underlying warm sea water, causes a cyclone surface wind field anomaly. Consequently, poleward heat transport enhances, and the presence of cyclone wind field heightens the likelihood of sea ice expansion. Panel (b) displays a vertical meridional view in EHI years, wherein extensive sea ice cover insulates air-sea heat exchange, inducing wind divergence that both facilitates sea ice accumulation and suppresses poleward heat transport.**

**4    Discussions**

**4.1    Coupling of SIA anomalies and wind fields in December**




Conventional understanding posits that large-scale SIA anomalies invariably induce significant SLP modifications. However, the sea ice variability analysed in this study is spatially concentrated on the north and south flanks of the Bering Strait, necessitating rigorous evaluation of whether such localized SIA changes can causally modulate SLP patterns. To systematically quantify the

causal linkages between December SIA and atmospheric dynamics, we employed two complementary approaches: the Liang-Kleeman information flow analysis and the PAMIP multi-model ensemble simulations. Contrary to expectations, the causal relationship between December SIA and SLP demonstrates complexity. As shown in Figure 8b and 8d, the SLP spatial patterns during LI and HI years exhibit substantial similarities, obscuring distinct sea ice distribution characteristics between these regimes. Through Liang-Kleeman information flow analysis applied to $SIA_{12}$ time series and the first three spatial patterns of SLP anomaly in December, we identified statistically insignificant causal relationship from $SIA_{12}$ to PC1 ($T_{SI\ 12 \to PC1}$ = 0.0043) and PC2

($T_{SI\ 12 \to PC}$ = 0.0013), falling below the 95% confidence level. In contrast, a significant information flow from $SIA_{12}$ to PC3 ($T_{SI\ 12 \to PC}$ = 0.0093, exceeding the 95% confidence level), indicating a notable causal influence of $SIA_{12}$ on EOF3, highlighting the potential importance of $SIA_{12}$ in shaping the SLP captured by EOF3. This selective causal relationship underscores the constrained capacity of $SIA_{12}$ in modulating December SLP patterns. Notably, while the influence of $SIA_{12}$ becomes detectable

during extreme sea ice events, the magnitude of its effect may remain eclipsed by dominant Arctic climate forcings (e.g., North Pacific SST anomalies, pan-Arctic sea ice variability, and thermal amplification feedbacks).

Of particular interest is the finding that the Liang-Kleeman information flow analysis reveals a statistically significant positive causal relationship between $SIA_{12}$ and wind divergence around St. Lawrence Island in Figure 11, which surpasses the 95% confidence level. Under extreme sea ice condition, the information flow exceeds 0.2, indicating a statistically robust causal linkage

(Figure 11b). This enhanced information flow demonstrates that December SIA variability exert influence through wind field modifications. While sea ice anomalies do not dominantly control the basin-scale spatial pattern of SLP, they modulate small-scale wind field changes via alterations in localized turbulent heat flux exchange. Notably, these mesoscale wind field adjustments act as a critical role in further regulating the substantial sea ice changes in the next month.

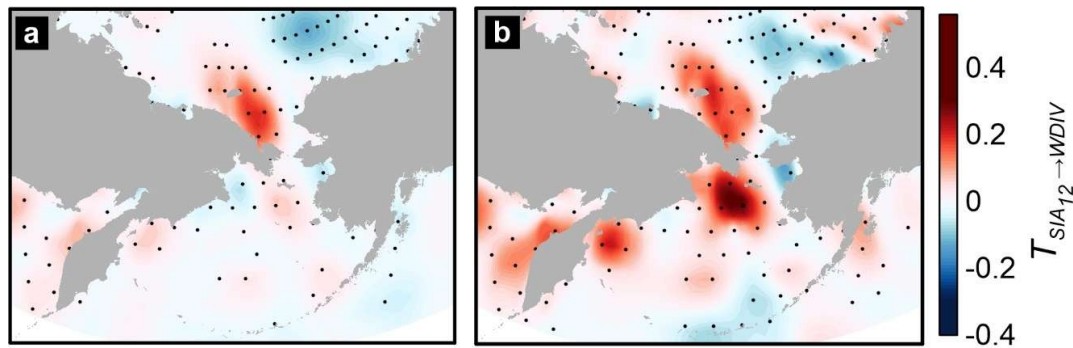

**Figure 11. Liang-Kleeman information flows from $SIA_{12}$ to December wind divergence (a). Panel (b) exclusively focuses on the causal relationship between the two variables under extreme sea ice conditions (EHI and ELI).**

The PAMIP simulations further corroborate pronounced differences in atmospheric feedback under contrasting sea ice forcing conditions, as quantified in Figure 12. During December, regions with extremely low SIC exhibit anomalous low-pressure systems

over the Bering Sea (Figure 12a), co-located with a distinctly cyclonic circulation anomaly. This pattern facilitates both enhanced sea ice divergence and intensified poleward thermal advection within the northern flank of cyclonic wind, collectively inhibiting equatorward sea ice expansion. Conversely, under extreme high-sea ice conditions, a substantial anticyclonic anomaly dominates





the west of Bering Sea (Figure 12b), generating surface wind divergence. Such a wind pattern, on the one hand, contributes to sea ice convergence, resulting in a reduced sea ice area. On the other hand, it suppresses northward heat transport, thereby favouring 430 sea ice expansion. While spatial discrepancies exist between simulated and observed SLP anomalies—notably the more poleward displacement of high-pressure anomalies in extreme heavy ice condition—both datasets consistently demonstrate a robust coupling between mesoscale atmospheric processes in December and extreme SIA variability around St. Lawrence Island.

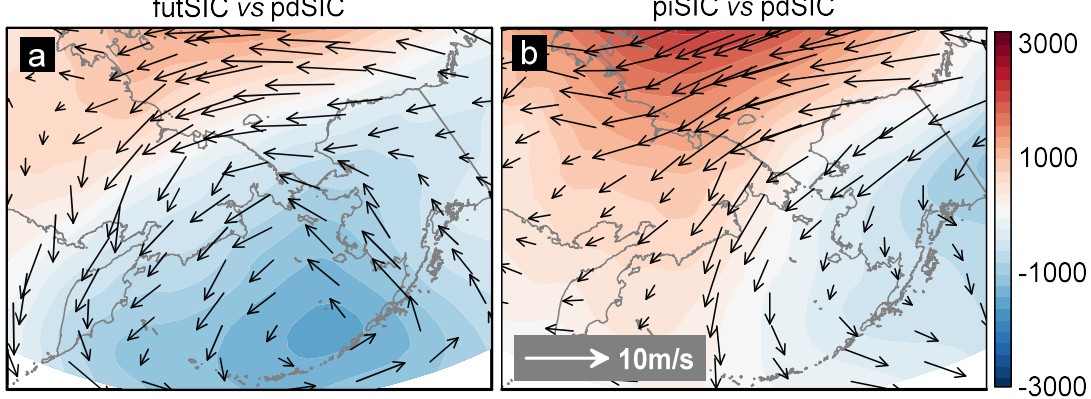

**Figure 12. Composite differences of SLP (unit: Pa, colour shading) and surface wind vectors (unit: m/s, arrows) derived**
**from CESM2 simulations under contrasting sea ice concentration (SIC) forcing scenarios. Panel (a) quantifies the SLP and wind vectors differences between future (futSIC) and present-day (pdSIC) forcings, whereas Panel (b) contrasts pre-industrial (piSIC) and pdSIC conditions.**

Rodionov et al. (2007) confirmed that the intensity and position of the AL consistently serve as significant proxies for pan-Arctic 440 climate variability and associated environmental shifts in the Northern Hemisphere. Extensive research has established linkages between AL variability and upper- tropospheric teleconnections (Overland et al., 1999; Trenberth and Hurrell, 1994), particularly the robust covariability between interannual AL intensity and the Pacific-North American teleconnection pattern (Lin et al., 2023; Sugimoto and Hanawa, 2009). Among known teleconnections, the Western Pacific (WP) pattern exhibits the strongest diagnostic capability for meridional AL displacements (Sugimoto and Hanawa, 2009; Wallace and Gutzler, 1981). Furthermore, the Arctic 445 Oscillation and El Niño Southern Oscillation also modulate AL's position and intensity (Gong et al., 2017; Trenberth and Hurrell, 1994). It is crucial to highlight that the impact of mesoscale sea ice changes on SLP described in the study represents a fine-structure variation of the AL. Whether these local changes affect the latitudinal movement and intensity of AL requires further detailed investigation.

Prior research endeavours have established that the spatial patterns identified in SLP during months extending beyond December 450 exhibit similarities to those characterized by the EOF3 derived from December SLP (Danielson et al., 2011; Iida et al., 2020; Rodionov et al., 2007; Stabeno et al., 2012b; Wendler et al., 2014; Zhang et al., 2010). These studies proposed that the northerly wind anomalies associated with this SLP pattern could facilitate southward advection of sea ice, thereby promoting SIA expansion. However, the findings presented in this study diverge from this perspective, particularly during early winter. Notably, Wang et al. (2022) demonstrate no statistically significant relationship between December wind speed and ΔSIA. The discrepancy with 455 conventional interpretations arises from the observation that in early winter, the response of local atmospheric circulation patterns – primarily reflected in local wind fields (encompassing both divergence and directional components) –initiates atmospheric and oceanic forcing mechanisms, which subsequently drive modifications in SIA during later month.





**4.2 Competition mechanisms between atmospheric and oceanic forcing**

Previous studies, including Li et al. (2014), Zhang et al. (2000, 2014), and Chen et al. (2014), concur that the interplay between
thermodynamic and dynamic forces governs the SIA in the Bering Sea. In light of the comprehensive analysis presented above,
the predominant driving factors influencing the first two spatial patterns of January ΔSIA in the Bering Sea are identified as (1)
northward heat transport at the sea ice edge in December (representing a thermodynamic process) and (2) wind divergence in
December (signifying a dynamic process). Therefore, we posit that the shift from interannual to decadal variability observed in the
January ΔSIA is the outcome of the intricate competition between atmospheric and oceanic forcing. The preceding findings suggest
an inherent antagonistic relationship between dynamic and thermodynamic forcings on sea ice in the Bering Sea, consistent with
the insights reported by Zhang *et al*. (2000). The impact of these dual driving factors on sea ice is modulated by the shared
background field (SIA$_{12}$). Thus, to elucidate the competition process accurately, it becomes imperative to directly compare the
magnitudes of these two distinct factors.

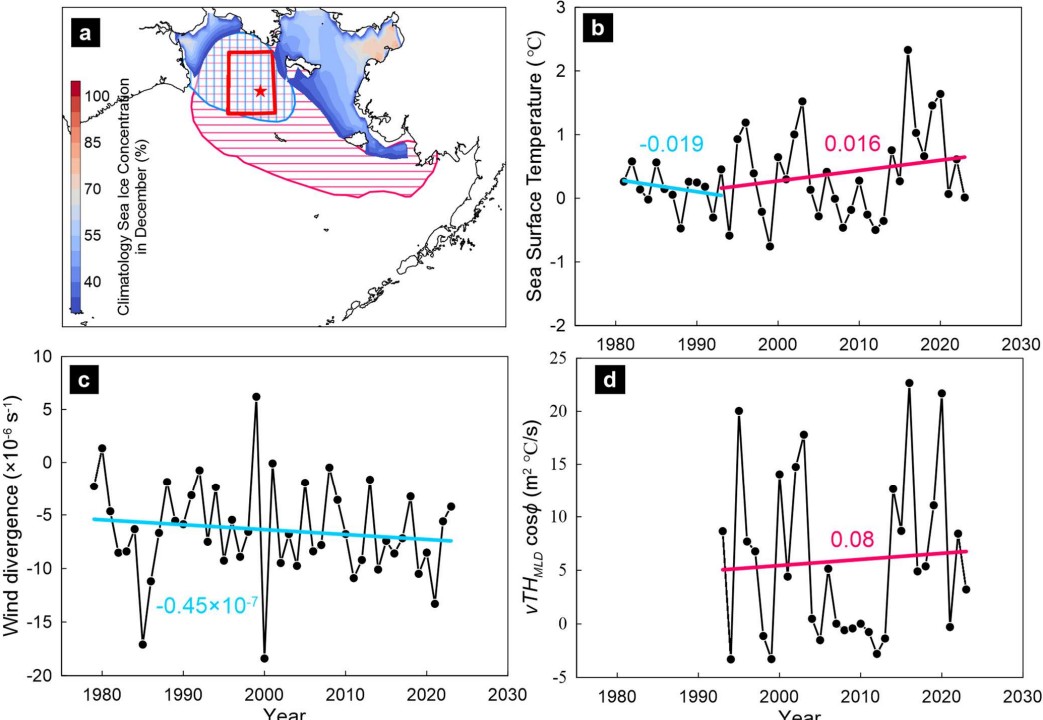

**Figure 13. The time series of the local December SST (b), wind divergence (c) and $VTH_{MLD}\cos\phi$ (d) from 1979 to 2023**
**surrounding the coordinates 62.40°N, -174.2°W (red star) denoted by red rectangle in Panel (a). The blue line in Panel (b,**
**c) denotes a negative trend, indicating a gradual decrease in the variable of interest over time. Conversely, the red line in**
**Panel (b, d) signifies a positive trend, demonstrating an increase in the variable. These trends do not meet the 95%**
**confidence level.**


As the northward heat transport is primarily governed by wind-driven Ekman transport, both driving factors—wind divergence
and northward heat transport—share a common physical driver: the wind vector. Additionally, northward heat transport is





modulated by SST. A detailed examination of the regions with significant correlation coefficients, as depicted in Figure 9, identifies the southern Bering Sea shelf as the focal area where these driving factors exert the most pronounced effect on sea ice (Figure 13a).

Consequently, we exacted the time series of the local SST, wind divergence and northward heat transport (quantified as $\varrho = VTH_{MLD} \cos \phi$) from 1979 to 2023 within the shared region, as shown in Figure 13. The observed SST (Figure 13b) decreased markedly at a rate of -0.019°C/year from 1979 to 1993, followed by a sustained increase of 0.016°C/year after 1994. Concurrently, $\varrho$ exhibited a upward trend since 1993, with a value of 0.08 m$^2$ °C/(s·year), indicating intensified oceanic forcing. In contrast, atmospheric forcing—represented by wind field divergence (Figure 13c) —has declined consistently since 1979, suggesting an

enhanced capacity of wind fields to promote SIA expansion. These divergent trends between oceanic and atmospheric forcings provide a mechanistic framework for interpreting the shift in ΔSIA variability: the transition from interannual to decadal fluctuations post-1994 aligns temporally with the 20-year scale intensification of northward warm transport.

Figure 14 schematically illustrates the mechanism governing the timescale transition of January ΔSIA. The process initiates with December SIA anomalies, which triggers two distinct feedback mechanisms during the same month: (1) positive feedback mediated

by oceanic heat transport, driving in-phase variability between December SIA and subsequent January ΔSIA, and (2) negative feedback governed by atmospheric forcing (wind divergence), generating antiphase changes in January ΔSIA. the temporal dominance of these forcings determines the characteristic timescale of ΔSIA variability. During 1979–1994, when atmospheric forcing prevailed, ΔSIA exhibited predominantly interannual fluctuations (Figure 13c). Since 1994, the growing influence of oceanic forcing has driven a transition to decadal-scale variability, persisting for approximately 30 years (Figure 13b).

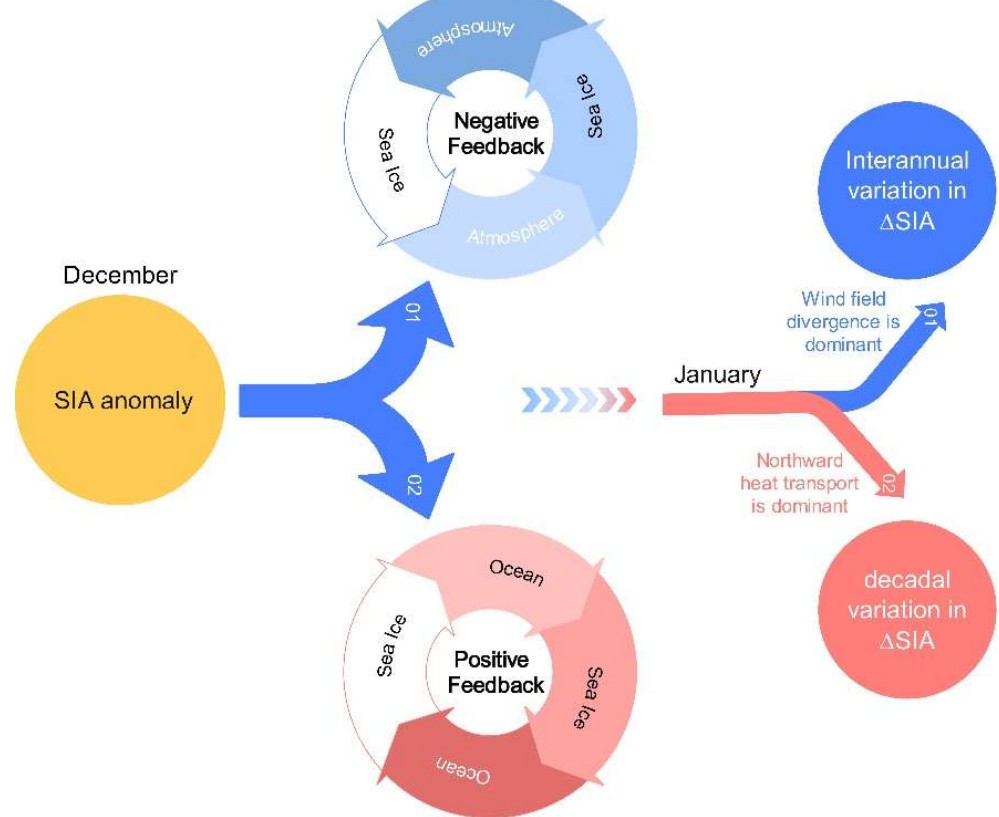





**Figure 14. Possible processes contributing to the timescale transition from interannual to decadal of SIA. The December SIA anomaly triggers two feedback processes: oceanic feedback and atmospheric feedback. Oceanic feedback manifests as a positive feedback mechanism, inducing decadal fluctuations in ΔSIA. Conversely, negative feedback from the atmosphere yields interannual fluctuations in sea ice. The competition between wind field divergence and northward heat**
**transport determining the fluctuation pattern of ΔSIA in January.**

Post-2018, a gradual recovery in the December SIA has been observed (Figure 2a). Concurrently, the ΔSIA in January has returned to a positive anomaly, with the ΔSIA in 2023 reaching $2.7 \times 10^5$ km²—ranking sixth in the 45-year satellite record. As a result of these changes, the maximum SIA in 2023 is approaching the historical average level (Figure 2e). Preliminary 2024 data indicate
further recovery, with the maximum SIA reaching $5.1 \times 10^5$ km²—very close to the historical average level. Given that it has been 10 years since the rapid decline of sea ice in 2013, the recent return to a positive anomaly warrants close attention to determine whether this trend signifies a potential continuation of positive sea ice anomalies in the Bering Sea in the future.

## 5    Conclusions

While atmospheric and oceanic forcings are recognized as primary drivers of sea ice variability in the Bering Sea, their specific
pathways and initiation triggers remain elusive. This knowledge gap persists despite observed regime shifts in SIA temporal characteristics—specifically, the transition from interannual to decadal dominance since the mid-1990s. This study identifies a pivotal observation regarding the earliest manifestation of timescale transition in SIA variability occurring in January. To explain this phenomenon, we propose a dual-feedback mechanism driven by oceanic and atmospheric forcing. Specifically, December SIA anomaly triggers two opposing processes: (1) positive feedback mediated by enhanced northward heat transport, which amplifies
multi-year and decadal SIA variability; (2) negative feedback mechanism controlled by wind-field divergence, primarily governing interannual ice fluctuations. Since 1994, the enhanced role of oceanic heat transport in modulating sea ice variability has amplified a positive ice-ocean feedback loop. This shift explains why oceanic forcing now dominates decadal-scale sea ice variability in the Bering Sea, supplanting the previously dominant atmospheric drivers that characterized the late 20th century.

This study demonstrates that the atmospheric response to sea ice changes is predominantly localized in regions experiencing
significant sea ice anomalies. Although the impact of sea ice on the large-scale atmospheric circulation patterns appears to be limited, these localized effects can substantially influence subsequent sea ice changes, potentially initiating cascading atmospheric responses that ultimately affect the mid-to-high latitude climate system. Recent research has identified a robust correlation between the PC1 in January ΔSIA and the maximum SIA (Wang et al., 2022). Future research should prioritize investigating this sea ice-driven causal chain to elucidate the tripartite interactions among sea ice, atmospheric processes, and oceanic dynamics.

The ecological and hydrological consequences of long-term heavy/light ice in the Bering Sea are multifaceted, including the disappearance of the cold pool in the eastern Bering Shelf and the poleward migration of subarctic groundfish communities. Ecological species in the Bering Sea have shifted from interannual to decadal fluctuation, impacting commercial and subsistence fishing production. This transformation poses challenges for indigenous communities and commercial enterprises reliant on fishery productivity. In the face of global warming, the persistent warming of seawater suggests that the sea ice cover in Bering Sea is
unlikely to revert to interannual changes. Instead, it appears that the Bering Sea may sustain prolonged periods of heavy or light ice, signalling a shift in fishery resource fluctuations from interannual to decadal timescale. Consequently, communities dependent on fishery resources must adapt to this more stable decadal-term environment. Simultaneously, they may encounter challenges stemming from substantial alterations in fishing production occurring at ten-year intervals.



**Data availability**

The monthly mean atmospheric variables are from NCEP/DOE AMIP-II reanalysis datasets at a 2.5°×2.5° spatial resolution
(https://www.psl.noaa.gov/data/gridded/data.ncep.reanalysis2.html) . We also use the monthly mean sea surface temperature
from the National Oceanic and Atmospheric Administration (NOAA) optimum interpolation SST (OISST) product on a
0.25°×0.25° grid (https://www.ncei.noaa.gov/products/optimum-interpolation-sst) , sea surface height on a 0.25°×0.25° grid

(https://data.marine.copernicus.eu/product/SEALEVEL_GLO_PHY_L4_MY_008_047/description), and sea ice concentration in
the polar stereographic projection at a grid cell size of 25 x 25 km (https://nsidc.org/data/nsidc-0051/versions/2). Data on the sea
ice concentration and atmospheric variables in the PAMIP experiments are obtained from Coupled Model Intercomparison
Project Phase Six (CMIP6) (https://esgf-data.dkrz.de/projects/cmip6-dkrz/) .

**Code availability**

All the codes used here are available from the corresponding author on reasonable request.

**Acknowledgements**

This work was supported by the Scientific Research Foundation of Third Institute of Oceanography, MNR (Grant 2024023), Fujian

Provincial Natural Science Foundation (General Project) (Grant 2024J01183), Special Fund for Marine Services and High-Quality
Development of Fisheries in Fujian Province (Grant FJHY-YYKJ-2024-1-12), National Key R&D Program of China (Grant
2024YFC28157X), and the National Natural Science Foundation of China (No. 42130406).

**Author contributions**

W.W. wrote the initial paper and carried out most of the data analysis. C. J., J. Z. and X. G. helped in the analysis of the data, and
revised the paper. W.W., C. J. and X. G. checked the paper and proposed amendments. All authors contributed to the paper and
approved the submitted version.

**Competing interests**

The authors declare no competing financial or non-financial interests.

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
