# Peer review of "Recent timescale transition from interannual to decadal variability in January sea ice area over the Bering Sea"

_EGUsphere, 2025_

## Referee Comment (RC2)

The change in Arctic sea ice area is a hot topic. Authors elucidated the changes in sea ice area (SIA) in boreal winter and amplified variance of decadal fluctuations in the wintertime SIA increment (ΔSIA).The manuscript is well-written and has good novelty. Some concerns should be addressed before acceptance of the manuscript I suggest minor revisions. Some grammar errors need to be revised. My detailed comments are organized as follows:

1. Sea ice concentration budget holds significant potential for elucidating the aforementioned competitive processes, as it enables the quantitative decomposition of sea ice concentration changes into distinct physical components—including sea ice drift (encompassing transport and convergence dynamics), thermal melting/freezing, and mechanical redistribution. Given this analytical capability, the authors are encouraged to clarify why the sea ice concentration budget was not employed in the present study.

2. What is the role of atmospheric thermodynamic forcing—such as the radiative effects of water vapor and clouds—in the proposed competition mechanism? These factors are known to exert significant influences on sea ice thermodynamic processes.

3. Section 2.2b: Would alternative decomposition methods (e.g., Ensemble Empirical Mode Decomposition, EEMD) produce similar results to the one currently used? Validating the findings against EEMD would help confirm the robustness of the analysis.

4. Figure 2i 'increament'->'increment'.

5. Figure 3: Have you conducted sensitivity tests on the moving window parameters? The robustness of the results presented in this figure relies heavily on the stability of the moving window settings.

6. Certain figures require additional beautification to improve their presentation quality, including Figures 4, 5, and 7.

7. Authors should emphasize further the implication of this study.

---

## Author Comment (AC1)

Dear reviewer,

Following a thorough review of your comments, we confirm that your suggestions will substantially enhance the overall quality of our manuscript. Subsequently, we will implement point-by-point revisions in accordance with the feedback provided. We sincerely appreciate your insightful and constructive feedback.

This article presents an exploration of the processes underlying a documented recent timescale transition in Bering Sea ice area from interannual to decadal variability. The mechanisms, as suggested by the authors, are a negative feedback wherein wind field anomalies resulting from sea ice area anomalies (due to localized atmosphere-ocean heat flux changes) induce changes in wind field divergence which impact subsequent sea ice area, and a positive feedback wherein wind field anomalies resulting from sea ice area anomalies impact oceanic heat transport which then impacts subsequent sea ice area. A change in the relative prevalence of these processes over time is suggested as an explanation for the transition from interannual to decadal variability.

Weibo: Thank you. This is precisely the message we aim to convey to our readership.

The ideas presented are scientifically interesting and the overall structure of the article is reasonably good. The quality of the figures is good overall. I think some of the points in Section 3 could be de-emphasized as I discuss below, for clarity and brevity, since it appears to me that the more substantial points are in Section 4. The dynamical/thermodynamical reasoning in the manuscript is reasonable to me, but I have some questions about the robustness of some of the arguments used to substantiate this reasoning. Furthermore, I found it was often difficult to ascertain in the article where there are novel findings and where findings are being cited/reproduced from previous work to provide context for the article's findings.

Because of this, I think the manuscript would require some substantial revisions before being suitable for publication.

Weibo: We greatly appreciate your positive assessment of the overall manuscript. We acknowledge that, to enhance the manuscript's readability, we have incorporated results from our prior work (Wang et al., 2023, Frontiers in Marine Science), including re-presented figures. In the revised version, we will segregate this content and place greater emphasis on the novel findings presented herein.

- Wang, W., Su, J., Jing, C., Guo, X., 2022. The inhibition of warm advection on the southward expansion of sea ice during early winter in the Bering Sea. Front. Mar. Sci. 9, 1–14. https://doi.org/10.3389/fmars.2022.946824

My general comments are included below, followed by line-by-line comments and technical corrections.

General comments:

A portion of this manuscript is closely connected to results from previous work by the authors; e.g. Wang et al. 2022, Wang et al. 2023, and Wang et al. 2024 a,b (as cited in the article references). Some of the panels in the figures are also found in previous work, and although the studies are referenced, it is not always clear which figure panels contain new work. I appreciate that the authors want to build up context from previous work to situate the current findings, but this makes it difficult for me to ascertain where the novel results in this article are located. I will try to indicate in my line-by-line comments where I see this happening but overall, I think many parts of Section 3 need revision to make it clearer where figure panels and results are reproduced from previous work.

Weibo: The manuscript does reference our prior research. In the revised version, we will address this issue by emphasizing the novel findings. Some of the reprocessed figures will be either removed or relocated to the appendix, with the associated text modified accordingly.

In general, parts of Sections 3.1 and 3.2 appear to be somewhat redundant to me; I think they could be condensed or partly moved to supplements. Because the timescale transition has been documented in previous work, I do not think as much detail is necessary in establishing the context except for the specific novel results on the specific timing of the transition, as applicable.

Weibo: Indeed, Sections 3.1 and 3.2 provide extensive detail regarding the time-scale transition. In response to your comments, we will address this issue in the revised manuscript. To optimize the balance of detail for readability, I propose relocating a portion of the content from Section 3.2 to an appendix.

The authors categorize December SIA years into five categories based on normalized values which is a reasonable approach, but I wonder at how representative the heavy-ice year composites are, since only ~5 years are included. Since much of the analysis seems focused on the extremes, I wonder if the authors could consider reducing the number of categories to simplify interpretation and streamline the results (i.e. reduce to 3 categories, either combine the low/high categories into the normal category, or group them with the extremes).

Weibo: We acknowledge that a sample size of five heavy ice years (HI) is limited for robust composite analysis. Furthermore, our subsequent analyses indicate that the atmospheric and oceanic impacts of both heavy ice years (HI) and light ice years (LI) are relatively modest. Therefore, as suggested by the reviewer, we will revise the manuscript to reduce the classification to three categories by incorporating these years into the normal year group.

The information flow argument for causal influence is broadly reasonable to me; given that the SLP captured by EOF3 appears to align with the SLP patterns during extreme high and low sea ice years, the statistically significant causal influence of SIA12 on EOF3 makes sense to me, although this influence appears to have a somewhat small magnitude. The relationship to wind divergence during extreme events is also fairly convincing. I have one question on this point: why is the same

causality analysis not carried out for other quantities such as SST and heat transport? Would we not expect them to also be causally linked to SIA12?

Weibo: We appreciate your positive assessment of the causal analysis. In the revised manuscript, we will incorporate causality analysis for heat transport and sea surface temperature (SST) with sea ice area (SIA12).

The reasoning around Figure 13 seems tenuous to me. The authors present time series of sea surface temperature, wind divergence, and heat transport in Figure 13, and discuss the trends in these quantities, using the differing trends to provide an interpretation for the change from interannual to decadal variability due to the dominance of differing contributing factors over the years. The physical explanation does seem reasonable to me, and earlier analysis in the manuscript does seem to substantiate it. However, the use of trends in Figure 13 weakens the argument for me because the trends are quite small and furthermore, they do not meet the 95% confidence level (I assume due to the high interannual variability). This is particularly apparent to me for the SST, where the authors report a decline followed by an increase as per the trend lines drawn in the plot, but equally, I could see a claim to be made for a decline from 1980 to ~2013, followed by a later increase, or also a relative lack of trend overall from 1979-2023, depending on where one would choose to place the discontinuity in trend lines. The heat transport is also highly variable and could also be interpreted to e.g. have a declining trend since ~2014. I would be more convinced if the authors could show that the relative magnitude of the driving factors changes, of which I am not convinced from Fig 13. I would appreciate if the authors could justify their inclusion of this analysis or make the argument using more robust data.

Weibo: We acknowledge the validity of your comment. We recognize that attributing the time-scale shift to long-term trends represents an erroneous interpretation in the original manuscript. The primary driver underlying the observed time-scale transition is, in fact, the magnitude of atmospheric and oceanic anomalies.

As illustrated in Figure 13, the sea surface temperature (SST) exhibits a transition from interannual to interdecadal variability, characterized by a gradual intensification of anomalous fluctuations. However, with the exception of several years distinguished by strong anomalies in wind field divergence, pronounced variability is not evident during other periods.

With the discussion of horizontal sea ice convergence/divergence, a consideration that is missing to me is sea ice thickness. With sea ice convergence comes thicker ice which can have thermodynamic and dynamic impacts, e.g. stronger inhibition of air-sea heat fluxes and impacts on ice motion. I would appreciate hearing at least a brief mention of what impacts the authors could expect from thicker ice.

Weibo: Indeed, sea ice thickness exerts a substantial influence on both air-sea fluxes and sea ice dynamics. However, owing to the paucity of available sea ice thickness data, particularly within the Bering Sea region, we are currently unable to incorporate this factor into our analysis. In response to the reviewer's comment, we will include a brief discussion of sea ice thickness in the revised manuscript's Discussion section.

I will leave this point to the authors' discretion, but I would find it helpful to have a more specific title (e.g. mention that the article is examining the role of mesoscale processes in the timescale transition).

Weibo: We will revise the manuscript's title in accordance with your suggestion.

Specific comments:

- L105: Why was ERA5 relegated to the supplement rather than NCEP? To my understanding it's a more contemporary dataset than the NCEP product with a higher resolution, which is particularly helpful when considering processes such as wind where capturing small scales is important

Weibo: During the initial draft preparation in 2024, our analysis primarily relied on NCEP 2 data. Subsequent to manuscript submission, reviewers recommended

supplementing the analysis with ERA5 data to validate the robustness of our findings. Accordingly, we incorporated the ERA5-derived results into an appendix. Notably, the outcomes derived from both datasets exhibit fundamental consistency. Consequently, no further modifications to the core analysis were implemented.

- L106: Reanalysis products are not observations, please take care with not referring to them as such here and throughout the article

Weibo: We will revise this issue according to your comment.

- L188: Please cite the "several studies" that have substantiated this transition

Weibo: We will revise this issue according to your comment.

- L190: The motivation for the use of IMFs here is not entirely clear to me; has the onset of the transition not been documented sufficiently in previous work?

Weibo: The primary rationale for employing IMFs to characterize the timescale transition of Bering Sea SIA is twofold. First, no prior studies have identified the existence of a timescale transition process in Bering Sea SIA; second, this analytical approach allows us to investigate the specific month in which such a timescale transition initiate. Our analyses reveal that this transition first emerges in January, which constitutes the core motivation for the present study's focus on the January SIA increment. In the revised manuscript, we will implement the requisite revisions as follows: (1) explicitly integrating the research objectives of this study within the Introduction section; (2) elaborating on the methodological rationale for the application of IMFs in the Method section; and (3) presenting a more precise and detailed characterization of the novel findings derived from IMFs in Section 3.1.

- L198-200/Figure S5: it's unclear to me how this figure illustrates the timescale transition mentioned in the text, since those figures are just time series, and they do not include any sort of analysis of variability. If this is a result shown

more clearly in previous work, I request that the authors state this more explicitly.

Weibo: These figures were initially intended to demonstrate that the timescale transition occurs in January. However, we consider that removing this content will enhance the clarity of the presentation. Accordingly, these figures will be removed in the revised manuscript.

- Figure S2, re: the wavelet transform: how reliable can some of these results be given that the edge effect influence extends quite far into the time series?

Weibo: We acknowledge that boundary effects are inevitably encountered during wavelet analysis, as such effects are inherently unavoidable in this analytical approach. This constitutes the primary rationale for our employment of the MODWT method and moving-window correlation test in the main text, while relegating the more intuitively interpretable wavelet analysis results to the appendix.

In response to your comment, we will add explicit annotations regarding the impact of edge effects to Figure S2 in the revised manuscript.

- Figure 4: Could you explain why the MODWT is unable to separate multiyear and decadal variability? Is it a matter of the time series not being long enough?

Weibo: We sincerely appreciate your insightful comments. Herein, the MODWT method employed enables the explicit capture of multiyear and decadal variability; yet for the sake of streamlined presentation, these two components were aggregated in the illustration of our results. This analytical treatment was adopted because our core objective was to demonstrate that interannual variability is weak in PC1 yet pronounced in PC2—a finding that constitutes the central message of the manuscript and the primary focus of our entire study. With respect to the multiyear and decadal variability embedded in PC1, no additional analysis was deemed necessary, as both components are inherently encapsulated within the PC1 signal.

- Figure 5 a): the coloured dots were initially somewhat confusing to me since they do not line up with the ice categories: consider changing them so that each category has a different colour or simply having all dots the same colour. The caption says 4 categories but there are 5.

Weibo: These revisions will be implemented in the revised version of the manuscript.

- L252-253: "only a single interannual variation feature is evident" could you clarify what you mean here? Is this a result from Wang et al. 2023?

Weibo: This statement is inaccurate and does not originate from Wang et al. (2023). We will revise this sentence accordingly in the revised manuscript.

- L260: Please rephrase, you say "significant" but the correlation not marked as statistically significant until after 2010 as per the green dots in Fig 5b

Weibo: This sentence will be revised in the revised manuscript to enhance its clarity and precision.

- L370: Maybe better to say the results are discussed further in Wang et al. 2022 rather than saying "corroborated" since that study is not entirely independent from this one (Figure 9b appears to have the same information as Fig 4b from that article)

Weibo: This sentence will be revised in the revised manuscript to enhance its clarity and precision.

- Figure 11: Please specify that dots indicate statistical significance in the caption

Weibo: I will revise this issue in the revised manuscript.

- L427/fig 12b: Minor phrasing consideration: the anticyclonic circulation only appears substantial north of the Bering Strait towards the East Siberian Sea

Weibo: I will revise this issue in the revised manuscript.

Weibo: The following technical corrections will be uniformly implemented in the revised version of the manuscript.

Technical corrections:

- L60: are -> is
- L131: "within annual" -> "within the annual"
- L295: "drivers" -> "drives"
- L307: anomalies.. -> anomalies.
- L311: I think this should be S3 not S1?
- L360: During -> during
- L379: influence -> influences
- L406: missing numbers for PC2 and PC3 in math text
- L415: exert -> exerts
- L480: exacted -> extracted?
- L491: the -> The